# Intimate partner violence and malnutrition among women of reproductive age in Western Africa: A geostatistical analysis

Ezra Gayawan[1]*, Endurance Uzobo[2], Dorothy N. Ononokpono[3], Olabimpe B. Aladeniyi[1], Fidelia A. A. Dake[4]

1 Department of Statistics, Federal University of Technology, Akure, Nigeria, 2 Department of Sociology, Niger Delta University, Wilberforce Island, Amassoma, Nigeria, 3 Department of Sociology and Anthropology, University of Uyo, Uyo, Nigeria, 4 Regional Institute for Population Studies, University of Ghana, Legon, Accra, Ghana

* egayawan@futa.edu.ng

**Data Availability Statement:** The data used for the study are freely available, upon request, from

## Abstract

Intimate partner violence (IPV) is a public health issue, and the experience varies among population sub-groups in Africa. In the West African sub-region, IPV perpetrated against women remains high and is exacerbated by the pertaining cultural milieu. It affects women's health, wellbeing, and nutritional status. We examined the association between women's lifetime experiences of physical, sexual, and emotional IPV and undernutrition by quantifying the association at smaller geographical settings in West African countries. We used a bivariate probit geostatistical technique to explore the association between IPV and undernutrition, combining data from the latest Demographic and Health Survey conducted in ten Western African countries. Bayesian inference relies on Markov chain Monte Carlo simulation. The findings demonstrate spatial clustering in the likelihood of experiencing IPV and being underweight in the regions of Mali, Sierra Leone, Liberia and neighboring Cote d'Ivoire, Ghana, Togo, Benin, Cameroon, and Nigeria. The pattern of clustering was somewhat similar when physical violence was combined with underweight and emotional violence combined with underweight. The findings also indicate protective effects of education, wealth status, employment status, urban residence, and exposure to mass media. Further, the likelihood of experiencing IPV and the likelihood of being underweight or thin declined with age and age-gap between the woman and her partner. The findings provide insight into the location-specific variations that can aid targeted interventions, and underscore the importance of empowering women holistically, in the domains of education, socio-economic and socio-cultural empowerment, in addressing women's vulnerability to IPV and malnutrition (underweight and thinness). Furthermore, IPV prevention programmes will need to address gender inequality and cultural factors such as male dominance that may heighten women's risk of experiencing IPV.

Demographic and Health Survey webpage at
https://dhsprogram.com/.

**Funding:** The author(s) received no specific
funding for this work.

**Competing interests:** The authors have declared
that no competing interests exist.

## Introduction

Intimate Partner Violence (IPV) is one of the most common forms of violence perpetrated against women by a current or former husband or intimate partner [1, 2]. The World Health Organization (WHO) defines IPV as any behavior within an intimate relationship, within the context of marriage, cohabitation or any formal or informal union [3], that leads to physical, mental, or sexual injury to others in the relationship, including physical harassment, sexual coercion, psychological torture, and controlling behaviors (WHO, 2002). IPV is a global public health issue and a violation of the rights of victims [4]. In 2018, the global estimates of lifetime prevalence of physical and or sexual IPV among women 15–49 years who have ever been married or partnered was highest in least developed countries at 37%, while Southern Asia and sub-Saharan Africa recorded 35% and 33% respectively [3, 5]. The prevalence of IPV among women in Africa continues to remain high even though most African countries have signed international declarations and developed several national laws to eliminate violence against women.

Several factors including sociocultural norms, beliefs and customs that favour men's hierarchical roles in sexual interactions, differences in age and education level, being married, socio-economic status of women, alcohol use by male partners, history of violence in both partners, and legal systems influence IPV among young women in sub-Saharan Africa [6–8]. The prevalence of IPV varies among the various sub-regions of Africa and among different population sub-groups. For instance, the Eastern and Southern Africa sub-regions have been noted to have the highest rate of sexual violence against women and girls [9] and data from seven Africa countries show that around 20% of the population aged 15 to 24 have been exposed to sexual and/or physical IPV at some point in their lives.

IPV has implications not only for the sexual and reproductive health of women but also their general health and wellbeing [10, 11] including their nutritional status. Evidence from research suggests that IPV is one of the main psychosocial factors that contributes to undernutrition especially among women [12]. Victims of IPV usually develop eating disorders, chronic fatigue and unhealthy weight loss which are all symptoms of undernutrition [13].

Previous studies conducted in Bangladesh and Nepal have identified impacts of IPV on malnutrition [14, 15], while previous studies in the sub-Saharan Africa context have focused primarily on the physical, sexual, and reproductive consequences of IPV, with limited attention given to its effects on malnutrition. But considering that IPV is context specific and in places such as West Africa where the prevalence of IPV is high and the prevailing socio-cultural environment including patriarchy and male dominance further exacerbates the perpetuation of IPV, it is important to unpack the relationship between IPV and malnutrition. Understanding the association between IPV and malnutrition in this region requires further investigation, as it could have significant implications for public health and social interventions aimed at addressing IPV and malnutrition. To the best of our knowledge, studies that have examined the relationship between IPV and malnutrition in the West African context are sparse. In this study, we seek to examine the relationship between IPV and malnutrition, specifically undernutrition (underweight and thinness) in the West African sub-region. We employ a bivariate probit geostatistical technique to jointly explore the association between IPV and undernutrition. We envisage that mapping and identifying countries in the region with high concentration of IPV and malnutrition is important for country-specific policies to address IPV and its adverse nutritional consequences on women of reproductive age.

## Methodology

### Data source

This study uses data from the Demographic and Health Survey (DHS) conducted in ten Western African countries. The DHS is a national sample survey that provides up-to-date information on population and health indicators to inform policy and provide data for planning, monitoring and evaluation of national health programs. The DHS usually follows a stratified two-stage cluster design. The first stage involves selecting sampling points (clusters) consisting of enumeration areas (EAs). EAs are drawn with probability proportional to their size within each sampling stratum. The second stage involves systematic sampling of households from a complete listing of households in each selected EAs. The survey interviews women of reproductive age (15–49 years) and men aged 15–59 years in the selected households. One eligible woman from each household in the sub-sample of participating households was randomly selected to be asked additional questions about domestic violence. The DHS surveys are conducted using similar instruments and methods across countries, the data are therefore comparable across countries and over time.

The study applied data from surveys conducted within the period of 2010 to 2020 in the following countries: Mali (2018), Nigeria (2018), Sierra Leone (2019), Liberia (2019–2020), Cote d'Ivoire (2011–2012), Burkina Faso (2010), Ghana (2014), Togo (2013–2014), Benin (2017–2018) and Cameroon (2018). It is assumed that period effect on the data within the ten year interval would be minimal. The following explanatory variables were considered: women's level of education, household wealth index, type of place of residence, woman's working status, access to mass media (whether or not the women accessed radio, television and newspaper/magazine at least once in a week), age of the women and the age gap between the woman and her partner. Data on the nutritional status of the women measured using their body mass index (BMI), which was calculated using the weight and height of the women was also analysed. In this study, a woman is said to be underweight if her BMI is less than 18.5 kg/m$^2$ and a woman whose BMI is less than 17 kg/m$^2$ is considered to be thin. Regarding IPV,—a women was considered to have experienced domestic violence if she has suffered physical, sexual or emotional violence, which were assessed through thirteen relevant questions, summarized in S1 Text). A positive answer to any one of the questions, whether ever or currently i.e. during the past year, suggested that the woman has experienced violence. In our application, we considered IPV with each of underweight and thinness, and further combined each of physical and emotional violence with underweight leaving out sexual violence because the variable has only few positive cases. The region/state or county where the woman lived at the time of the surveys were considered as the spatial unit.

### Methods of analysis

We considered a bi-probit spatial model because of its potential for modelling two correlated binary outcomes. As earlier argued, the two indicators of interest here: underweight and IPV can be highly correlated and each is measured as a binary indicator. The model incorporates a correlation parameter, ρ that accounts for the possible linear association between the indicators. Against the foregoing, let $Y_1$ be a binary indicator for the first outcome of interest, say IPV and $Y_2$ represents the second response, say underweight. The convention for a probit model is to assume that the response variables have been generated from some latent variables $Y_1^*$ and $Y_2^*$ respectively, satisfying the following conditions:

$$Y_1 = \begin{cases} 1 & \text{if} & Y_1^* > 0 \\ 0 & \text{if} & \text{otherwise} \end{cases}$$

and

$$Y_2 = \begin{cases} 1 & \text{if} & Y_2^* > 0 \\ 0 & \text{if} & \text{otherwise} \end{cases}.$$

Conditioned on available covariates of different types collectively denoted with the vector $\mathbf{v}$, the latent variables are assumed to jointly have a bivariate normal distribution denoted by:

$$\begin{bmatrix} Y_1^* \\ Y_2^* \end{bmatrix} | \mathbf{v} \sim N\left( \begin{bmatrix} \mu_1 \\ \mu_2 \end{bmatrix}, \begin{bmatrix} \sigma_1^2 & \rho\sigma_1\sigma_2 \\ \rho\sigma_1\sigma_2 & \sigma_2^2 \end{bmatrix} \right).$$

If the variance is assumed to be unity, we have $\begin{bmatrix} Y_1^* \\ Y_2^* \end{bmatrix} | \mathbf{v} \sim N(\mu, \sum)$ where $\mu' = (\mu_1, \mu_2)$ and $\sum$ is a 2×2 diagonal matrix with 1 as its major diagonal element and $\rho$, the correlation parameter for the two latent variables, as its off diagonal element. The three parameters collected together as $\vartheta' = (\mu_1, \mu_2, \rho)$ can be simultaneously linked to available covariates following the distributional regression framework through the predictor $\eta_i^{\vartheta_k}$ [16]. A link function is assigned to the predictor to allow for appropriate restrictions on the parameter space such that $\vartheta_k = h_k^{\vartheta_k}(\eta_i^{\vartheta_k})$ where $k = 1,2,3$, are the number of parameters to be linked and $i$ is the number of children. The mean parameters, $\mu_1$ and $\mu_2$ are each assigned an identity link function while the correlation parameter, $\rho$ was assigned a Fisher Z-transformation link function. The generic predictor is then related to the available covariates as follows:

$$\eta_i^{\vartheta_k} = \alpha^{\vartheta_k} + \beta^{\vartheta_k}(v_i^m) + f_p^{\vartheta_k}(v_i^{(l)}) + u^{\vartheta_k}(s_i)$$

where the additive terms are defined as follows: $\alpha^{\vartheta_k}$ is the model intercept for the $k^{th}$ parameter, $\beta^{\vartheta_k}$ is a vector for the linear parameters for the categorical covariates $v_i^m$, $f_p^{\vartheta_k}$ are smooth, non-linear functions assumed for the metrical covariates $v_i^{(l)}$ such as the age of the woman and age gap, and $u^{\vartheta_k}$ are spatial-structured offset for each of the distributional parameters. Because of the complexity of the model parameters that involve covariates of different types, a Bayesian approach is a suitable approach for parameter estimation, because of the suitability of sampling from the posterior distribution which if often intractable. Consequently, prior distributions were assigned to the different model terms as follows: for the model intercept and linear parameters, a non-informative prior was considered. For the smooth, non-linear effects, we used a Bayesian P-spline prior [17, 18]. The approach allows for non-parametric estimation of the function as a linear combination of basis splines (B-splines). We used B-splines based on 20 equidistance knots and assigned a second-order random walk as hyperprior. Previous studies have shown that this choice is flexible enough to capture severe non-linearity [19, 20]. To model the structured spatial effects, the regions of the countries were considered as a discrete set of spatial locations $s_i \in \{1, \ldots, S\}$, and a Markov random field prior was assigned. The prior implements a binary structure for the neighbourhood relations of the regions such that regions that share a common boundary are assigned a weight of one and zero if they share no boundary. To enforce spatial smoothness, a Gaussian Markov random field prior that induces a penalty whereby spatially adjacent regions are penalized was employed. More details about possible prior choice and other functions that can be accommodated within the distributional regression framework can be found in [16, 21].

Fully Bayesian inference from the posterior distribution of the model parameters is based on the Markov chain Monte Carlo (MCMC) sampling techniques that generates samples from

the full conditionals for all parameters. We executed the MCMC sampler as a Metropolis-Hastings algorithm based on iteratively weighted least square (IWLS) as developed by [16] and implemented in BayesX version 3.0.2 –a freely distributed software package for Bayesian inference in structured additive regression models. We carried out the MCMC sampling based on a total of 35,000 iterations, setting the burn-in sample to be 5,000 and thinned every 30[th] observations for parameter estimation. We monitored the convergence of the samples through the plots of the sampling paths.

## Results

### Linear effects

Table 1 presents the estimates of the linear parameters for IPV and underweight, revealing that women who attained primary or secondary level of education have higher chances of experiencing IPV compared to those with no education but those with higher level of education are less likely. Women from the richer and richest households have lower chances of experiencing IPV. Also, the chances of experiencing IPV are higher for those who were working compared to those who are not working. For access to media, those who listen to radio at least once in a week are at higher risk of experiencing IPV. The other estimates for IPV were not significant (Table 1).The results for underweight show that compared with women who had no education, women with at least primary education were less likely to be underweight.

**Table 1. Estimates of the linear parameters for IPV and underweight.**

| Variables | IPV | | Underweight | | Correlation | |
|---|---|---|---|---|---|---|
| | Posterior mean | Credible interval | Posterior mean | Credible interval | Posterior mean | Credible interval |
| Individual characteristics | | | | | | |
| Constant | -0.489 | -0.563, -0.418 | -1.169 | -1.274, -1.065 | 0.024 | -0.108, 0.163 |
| Place of residence | | | | | | |
| Rural | 0 | | 0 | | 0 | |
| Urban | 0.008 | -0.029, 0.045 | -0.058 | -0.115, 0.002 | -0.020 | -0.092, 0.050 |
| Level of education | | | | | | |
| No education | 0 | | 0 | | 0 | |
| Primary | 0.114 | 0.075, 0.153 | -0.131 | -0.194, -0.073 | 0.044 | -0.030, 0.121 |
| Secondary | 0.050 | 0.007, 0.093 | -0.080 | -0.150, -0.015 | -0.014 | -0.099, 0.064 |
| Higher | -0.249 | -0.334, -0.161 | -0.170 | -0.330, -0.005 | -0.158 | -0.404, 0.065 |
| Wealth index | | | | | | |
| Poorest | 0 | | 0 | | 0 | |
| Poorer | -0.021 | -0.064, 0.022 | -0.087 | -0.144, -0.031 | 0.057 | -0.02, 0.134 |
| Middle | -0.024 | -0.071, 0.021 | -0.186 | -0.247, -0.123 | -0.018 | -0.106, 0.072 |
| Richer | -0.082 | -0.136, -0.035 | -0.325 | -0.401, -0.250 | 0.041 | -0.063, 0.137 |
| Richest | -0.248 | -0.312, -0.182 | -0.520 | -0.620, -0.419 | 0.096 | -0.037, 0.22 |
| Employment status | | | | | | |
| Not working | 0 | | 0 | | 0 | |
| Working | 0.173 | 0.14, 0.208 | -0.070 | -0.122, -0.018 | -0.034 | -0.101, 0.036 |
| Access to media | | | | | | |
| No access | 0 | | 0 | | 0 | |
| Newspaper | -0.044 | -0.098, 0.012 | -0.051 | -0.146, 0.050 | -0.071 | -0.207, 0.062 |
| Radio | 0.059 | 0.031, 0.092 | -0.037 | -0.08, 0.009 | 0.043 | -0.017, 0.101 |
| Television | 0.024 | -0.013, 0.063 | -0.050 | -0.104, 0.005 | -0.010 | -0.086, 0.067 |

Similarly, women from at least the poorer households and those who were working at the time of the survey were less likely to be underweight.

The estimates for the correlation component indicate a negative linear relationship between IPV and underweight among women who lived in urban areas, those with secondary or higher education, those from middle wealth index households, those working, women who read newspaper and those who watch television (Table 1). The implication is that these categories of women were less likely to have simultaneously suffered from IPV and underweight. There was, however, a positive linear relationship for women who had primary education, those from the poorer, richer or richest households, and those who listen to radio. The results from the model that combines IPV and thinness presented in Table 2 are largely similar to those reported in Table 1 for IPV and underweight. The exceptions are that women who read newspaper have higher chances of being thin, while the linear correlation between IPV and being thin is positive for women who attained secondary education, those from middle wealth status households and working women but negative for those from poorer households.

The results for the models that individually combine physical and emotional violence with underweight are presented in Tables 3 and 4 respectively. The estimates for physical violence indicate higher chances for urban residency, attaining primary level of education, and being working but lower for higher level of education, and belonging to the richer or richest households. The estimates show positive association between physical violence and underweight among women from at least poorer households, those who listen to radio and those who watched television. For emotional violence (Table 4), the likelihoods are lower among women

**Table 2. Estimates of the linear parameters for IPV and thinness.**

| Variables in model | IPV | | Thin | | Correlation | |
|---|---|---|---|---|---|---|
| | Posterior mean | Credible interval | Posterior mean | Credible interval | Posterior mean | Credible interval |
| Individual characteristics | | | | | | |
| Constant | -0.486 | -0.559, -0.415 | -1.818 | -1.972, -1.67 | 0.177 | -0.042, 0.546 |
| Place of residence | | | | | | |
| Rural | 0 | | 0 | | 0 | |
| Urban | 0.006 | -0.031, 0.043 | -0.008 | -0.098, 0.080 | -0.037 | -0.157, 0.083 |
| Level of education | | | | | | |
| No education | 0 | | 0 | | 0 | |
| Primary | 0.114 | 0.075, 0.154 | -0.133 | -0.238, -0.036 | 0.062 | -0.07, 0.209 |
| Secondary | 0.052 | 0.008, 0. 96 | -0.113 | -0.227, -0.005 | 0.003 | -0.155, 0.153 |
| Higher | -0.246 | -0.33, -0.155 | -0.266 | -0.560, 0.005 | -0.029 | -0.578, 0.504 |
| Wealth index | | | | | | |
| Poorest | 0 | | 0 | | 0 | |
| Poorer | -0.022 | -0.064, 0.022 | -0.052 | -0.143, 0.044 | -0.058 | -0.188, 0.092 |
| Middle | -0.025 | -0.072, 0.022 | -0.068 | -0.164, 0.03 | 0.025 | -0.116, 0.166 |
| Richer | -0.082 | -0.135, -0.035 | -0.258 | -0.376, -0.135 | 0.005 | -0.157, 0.177 |
| Richest | -0.248 | -0.313, -0.183 | -0.429 | -0.594, -0.272 | 0.091 | -0.156, 0.331 |
| Working status | | | | | | |
| Not working | 0 | | 0 | | 0 | |
| Working | 0.174 | 0.14, 0.209 | -0.169 | -0.242, -0.09 | 0.011 | -0.092, 0.115 |
| Access to media | | | | | | |
| No access | 0 | | 0 | | 0 | |
| Newspaper | -0.044 | -0.098, 0.012 | 0.189 | 0.041, 0.334 | -0.176 | -0.402, 0.029 |
| Radio | 0.058 | 0.029, 0.091 | -0.058 | -0.129, 0.018 | 0.050 | -0.058, 0.153 |
| Television | 0.025 | -0.013, 0.064 | -0.071 | -0.163, 0.025 | -0.062 | -0.211, 0.072 |

**Table 3. Estimates of the linear parameters for physical violence and underweight.**

| Variables in model | Physical | | Underweight | | Correlation | |
|---|---|---|---|---|---|---|
| | Posterior mean | Credible interval | Posterior mean | Credible interval | Posterior mean | Credible interval |
| Individual characteristics | | | | | | |
| Constant | -0.857 | -0.939, -0.776 | -1.169 | -1.274,-1.068 | -0.029 | -0.165, 0.108 |
| Place of residence | | | | | | |
| Rural | 0 | | 0 | | 0 | |
| Urban | 0.045 | 0.004, 0.086 | -0.058 | -0.115, 0.001 | -0.031 | -0.108, 0.047 |
| Level of education | | | | | | |
| No education | 0 | | 0 | | 0 | |
| Primary | 0.132 | 0.091, 0.174 | -0.131 | -0.194, -0.072 | -0.027 | -0.11, 0.049 |
| Secondary | 0.044 | -0.003, 0.089 | -0.080 | -0.150, -0.015 | -0.029 | -0.126, 0.056 |
| Higher | -0.23 | -0.323, -0.133 | -0.167 | -0.331, -0.002 | -0.092 | -0.354, 0.167 |
| Wealth index | | | | | | |
| Poorest | 0 | | 0 | | 0 | |
| Poorer | -0.021 | -0.065, 0.026 | -0.086 | -0.143, -0.03 | 0.063 | -0.015, 0.148 |
| Middle | -0.03 | -0.08, 0.019 | -0.185 | -0.246, -0.122 | 0.024 | -0.069, 0.124 |
| Richer | -0.083 | -0.139, -0.031 | -0.324 | -0.399, -0.248 | 0.032 | -0.078, 0.134 |
| Richest | -0.256 | -0.328, -0.186 | -0.518 | -0.619, -0.416 | 0.131 | -0.014, 0.268 |
| Working status | | | | | | |
| Not working | 0 | | 0 | | 0 | |
| Working | 0.122 | 0.088, 0.159 | -0.072 | -0.124, -0.02 | -0.003 | -0.078, 0.068 |
| Access to media | | | | | | |
| No access | 0 | | 0 | | 0 | |
| Newspaper | -0.047 | -0.104, 0.013 | -0.049 | -0.144, 0.051 | -0.028 | -0.169, 0.11 |
| Radio | 0.014 | -0.016, 0.048 | -0.036 | -0.079, 0.01 | 0.037 | -0.029, 0.099 |
| Television | 0.026 | -0.014, 0.066 | -0.052 | -0.105, 0.004 | 0.009 | -0.073, 0.09 |

with higher level of education and those from the richer or richest households. However, women who with primary education, working women, and those who listen to radio have lower likelihoods. The estimates also show negative association for urban residency, secondary or higher education, poorer, middle and richest households, reading of newspaper and watching television but positive for primary education, and listening to radio.

## Spatial effects

The findings from the spatial effects are presented in Figs 1–4, showing the estimates for the mean levels (a & b) and the correlation parameter (c). The results demonstrate strong spatial clustering in IPV and undernutrition within and across the West African countries. Specifically, the likelihood that a woman would suffer IPV (Fig 1A) are higher among those who reside in Gombe, Bauchi, Plateau, Kaduna, Niger, Kogi, Adamawa, Taraba, Edo, Ebonyi, and Rivers states of Nigeria. This spans through neighbouring Cameroon covering the Centre (including Yaounde), East, South, Litoral, West, Northwest, Southwest, and Douala regions. Similarly, women who reside everywhere in Sierra Leone and Liberia; those from Kayes, Koulikoro, Segou, Sikasso, and Gao in Mali; the Western and Eastern regions of Ghana and most part of Benin have high likelihood of IPV but particularly lower likelihood for those living in Burkina Faso. The results also show higher of likelihood of underweight among women who reside in the northern fringe of Nigeria, throughout Burkina Faso; Kayes, Koulikoro, and Gao regions of Mali but lower likelihood for those living in Sierra Leone, Liberia and the southern

**Table 4. Estimates of the linear parameters for emotional violence and underweight.**

| Variables in model | Emotional | | Underweight | | Correlation | |
|---|---|---|---|---|---|---|
| | Posterior mean | Credible interval | Posterior mean | Credible interval | Posterior mean | Credible interval |
| Individual characteristics | | | | | | |
| Constant | -0.698 | -0.772, -0.625 | -1.17 | -1.273, -1.065 | 0.126 | -0.009, 0.267 |
| Place of residence | | | | | | |
| Rural | 0 | | 0 | | 0 | |
| Urban | -0.001 | -0.039, 0.037 | -0.057 | -0.115, 0.003 | -0.065 | -0.141, 0.011 |
| Level of education | | | | | | |
| No education | 0 | | 0 | | 0 | |
| Primary | 0.098 | 0.058, 0.139 | -0.131 | -0.194, -0.073 | 0.038 | -0.042, 0.113 |
| Secondary | 0.016 | -0.028, 0.06 | -0.081 | -0.151, -0.017 | -0.03 | -0.122, 0.049 |
| Higher | -0.273 | -0.361, -0.177 | -0.171 | -0.335, -0.007 | -0.144 | -0.41, 0.104 |
| Wealth index | | | | | | |
| Poorest | 0 | | 0 | | 0 | |
| Poorer | -0.008 | -0.051, 0.036 | -0.086 | -0.142, -0.03 | -0.005 | -0.084, 0.079 |
| Middle | -0.031 | -0.08, 0.016 | -0.184 | -0.245, -0.121 | -0.041 | -0.132, 0.047 |
| Richer | -0.089 | -0.142, -0.04 | -0.324 | -0.399, -0.248 | 0.057 | -0.049, 0.159 |
| Richest | -0.225 | -0.294, -0.158 | -0.519 | -0.619, -0.416 | -0.052 | -0.088, 0.191 |
| Working status | | | | | | |
| Not working | 0 | | 0 | | 0 | |
| Working | 0.157 | 0.122, 0.192 | -0.071 | -0.123, -0.018 | -0.074 | -0.144, -0.004 |
| Access to media | | | | | | |
| No access | 0 | | 0 | | 0 | |
| Newspaper | -0.04 | -0.096, 0.019 | -0.051 | -0.147,0.05 | -0.126 | -0.255, 0.015 |
| Radio | 0.067 | 0.037, 0.100 | -0.037 | -0.08, 0.009 | 0.056 | -0.007, 0.116 |
| Television | 0.021 | -0.019, 0.061 | -0.05 | -0.104, 0.005 | -0.028 | -0.108, 0.047 |

regions of Cameroon. The map for the correlation parameter (Fig 1C) shows a positive linear relationship between IPV and underweight among women residing in the Gao, Kidal, and Tombouctou regions of Mali; all parts of Sierra Leone and Liberia spanning through to the neighbouring Cote d'Ivoire, Ghana, and Togo but a negative among those residing in the Eastern part of Nigeria, extending to the northern regions of Cameroon. The estimates for IPV and thinness (Fig 2) show a higher likelihood of being thin among women living in the northern part of Nigeria and in Oyo in southern Nigeria; the northern regions of Cameroon; Kayes in Mali, and in the Sahel and Nord regions of Burkina Faso. The correlation only reflects negative linear relationship among the women living in Kayes of Mali and some southern part of Nigeria.

The maps from the model that combines physical violence and underweight (Fig 3) show that the chances of experiencing physical violence are higher among those living anywhere in Sierra Leone and Liberia; Kayes, Koulikoro (including Bamako), the Sikasso, and Segou regions of Mali; Northwest region of Cote d'Ivoire; Gombe, Niger, Edo, Ebonyi and Rivers States of Nigeria; and in the regions that make up the southern parts of Cameroon. But the likelihoods are lower among women residing in the northern fringe of Nigeria and in most regions of Burkina Faso. The correlations parameter show a positive relationship among the women living in the Gao, Kidal, Tombouctou, and Koulikoro regions of Mali and in Sierra Leone but negative for women living in most parts of Nigeria extending to Cameroon. The maps for emotional violence (Fig 4) show a higher likelihood among women living in Gombe, Bauchi, Plateau, Kaduna, Adamawa, Kogi, Edo, Delta and Anambra States of Nigeria; those living in Sierra Leone and Liberia; in the southern regions of Cameroon.

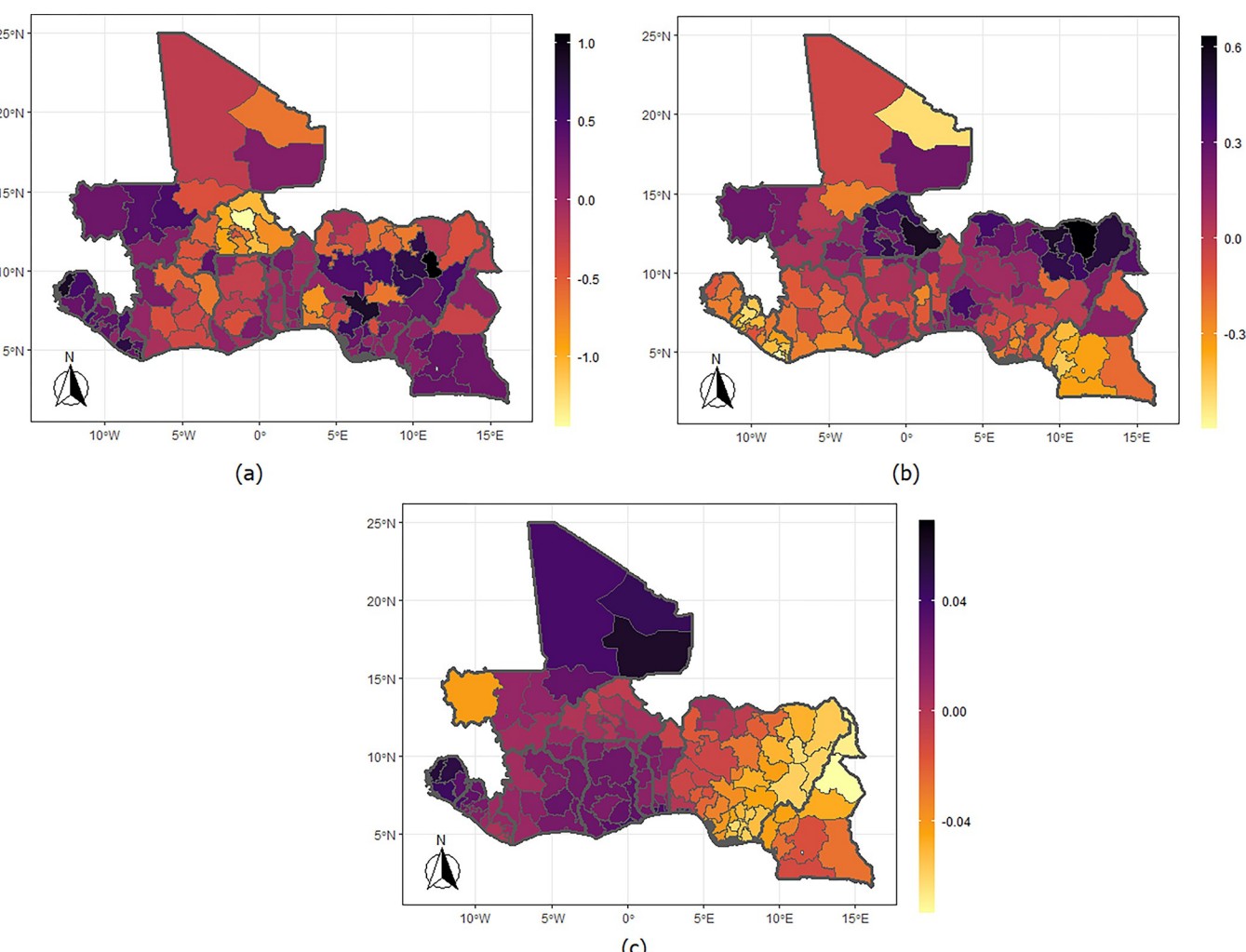

**Fig 1.** Maps of the West African countries showing the mean levels for (a) IPV, (b) underweight, and (c) correlation between IPV and underweight. Shapefiles used in plotting the maps were sourced from https://spatialdata.dhsprogram.com/home/.

## Non-linear effects

Figs 5–8 presents the plots for the estimates of the non-linear effects of the woman's age and age gap for IPV and underweight (Fig 5), IPV and thinness (Fig 6), physical violence and underweight (Fig 7), and emotional violence and underweight (Fig 8) for the means and correlation parameters. The Figures show the posterior means surrounded by the 95% credible intervals. For IPV, the findings indicate that the likelihood of experiencing IPV rises sharply with age to peak around 30 years from where it declines gradually. However, the likelihood of being underweight reduces sharply with age up to around 25 years followed by a gradual decline to around 38 years followed by a slight rise. The correlation parameter rises on the negative side, and slightly crosses the zero line around 30 years. The estimates for age gap (Fig 5, lower panel) indicate a somewhat linear relationship with IPV while the chances of underweight reduce with age gap, particularly when the gap widens beyond 30 years. The correlation parameter indicates a positive association between IPV and underweight when the woman is much older, but this quickly shrinks as the age difference collapses to around zero.

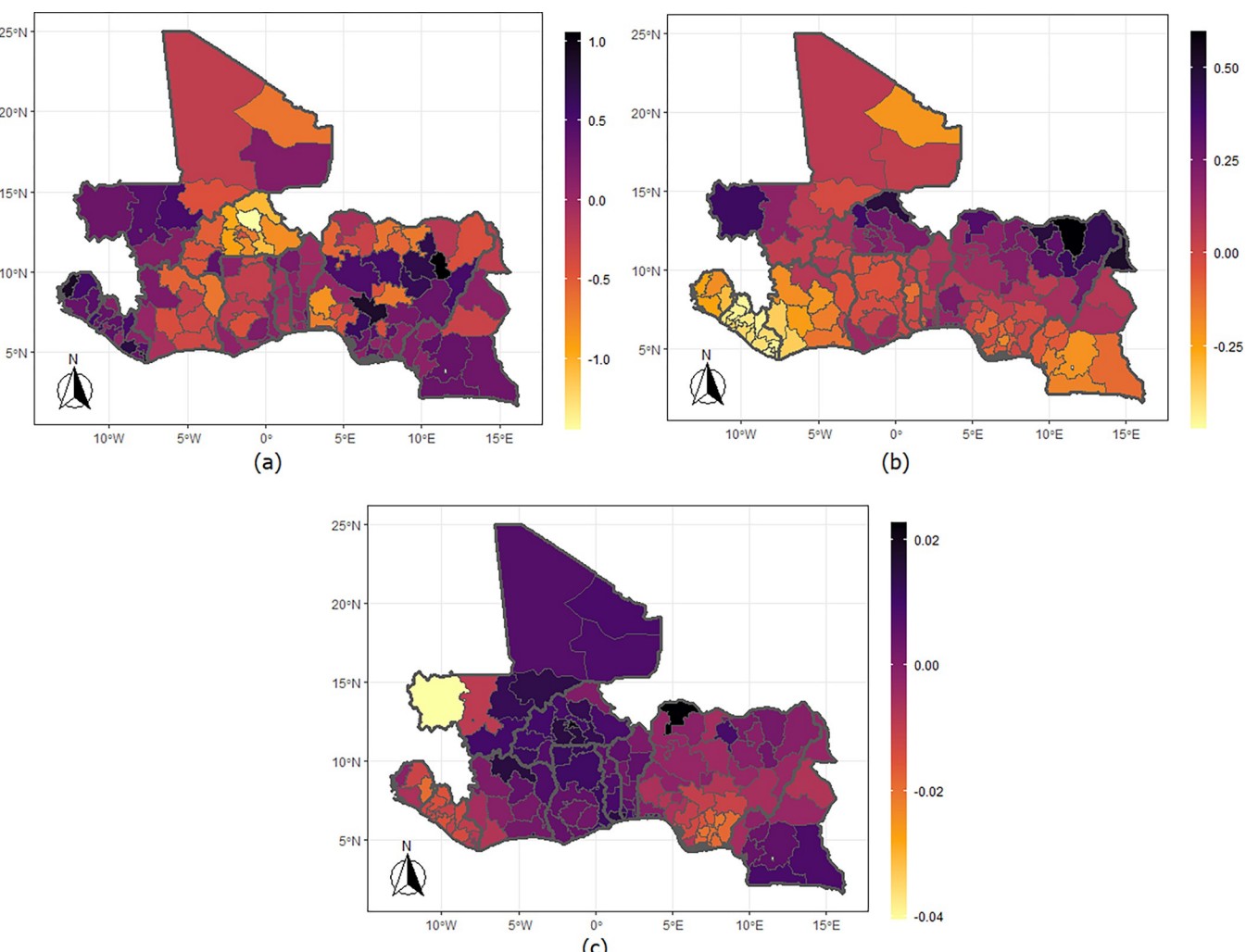

**Fig 2.** Maps of the West African countries showing the mean levels for (a) IPV, (b) thinness, and (c) correlation between IPV and thinness. Shapefiles used in plotting the maps were sourced from https://spatialdata.dhsprogram.com/home/.

The estimates from the model that combines IPV and thinness (Fig 6) shows that the likelihood that a woman would be thin is highest for women who were 15 years old but reduces drastically with age till around 28 years followed by another gradual rise. The estimates for the correlation parameter indicate a sinusoidal pattern that reduces after age 40 years. Regarding age gap, the findings for thinness are somewhat similar to those of underweight but the correlation rises with expansion in age gap. The results for the levels of physical violence and underweight (Fig 7) are similar to those of IPV and underweight. However, the correlation term reduces from 15 years to around 30 years and rises thereafter. The results for age gap are also similar except for the correlation that reveals a linear pattern around zero.

## Discussion

This study examined the association between IPV and malnutrition (underweight and thinness) among women of reproductive age in West Africa using a geo-statistical analysis approach. The findings reveal that there is an association between IPV and underweight controlling for other factors. The findings also indicate that there is spatial clustering in the co-

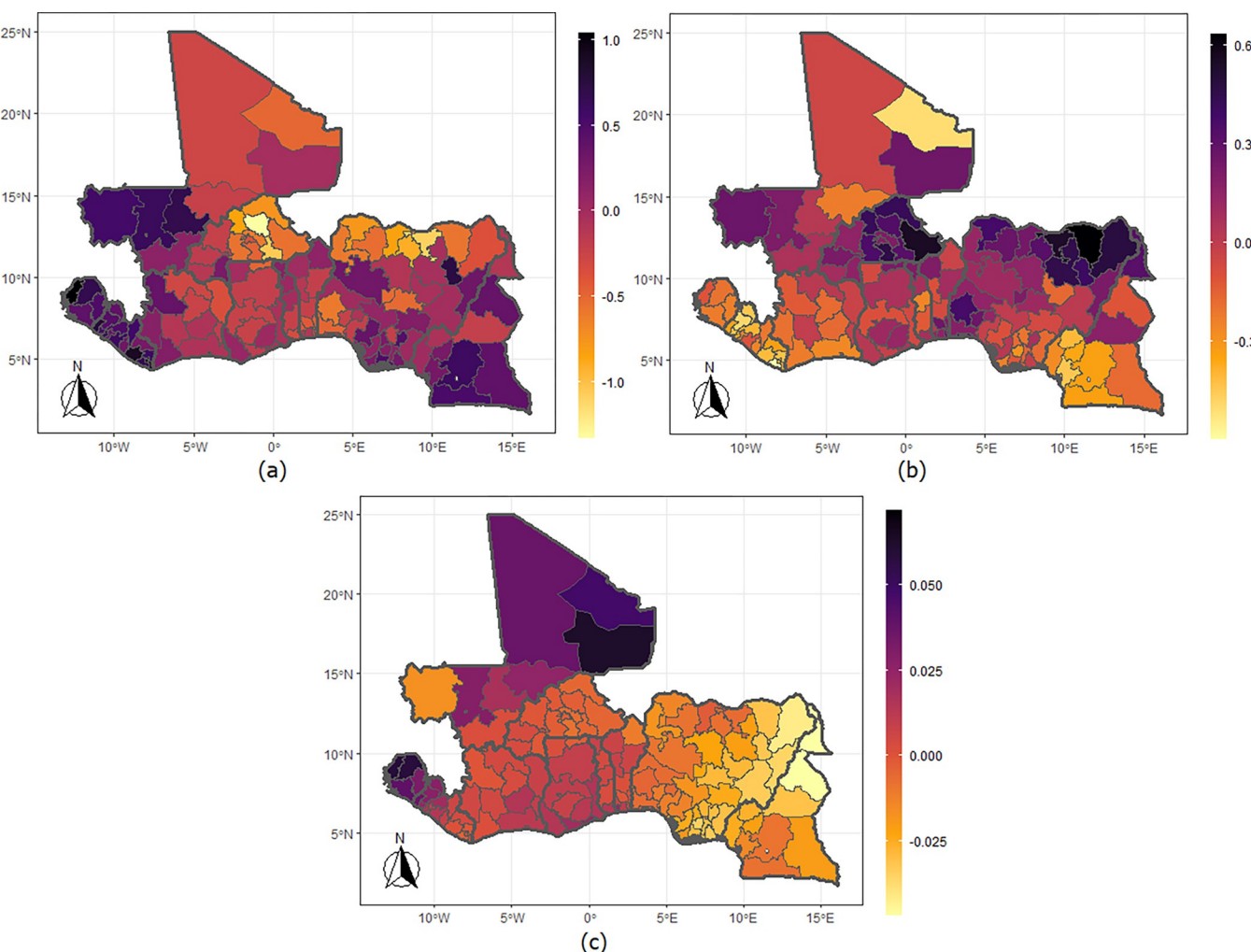

**Fig 3.** Maps of the West African countries showing the mean levels for (a) physical violence, (b) underweight, and (c) correlation between physical violence and underweight. Shapefiles used in plotting the maps were sourced from https://spatialdata.dhsprogram.com/home/.

occurrence of IPV and underweight among women of reproductive age across some geographic areas in the West African sub-region. Protective effects of education and wealth status against experiencing IPV and being underweight were found in the current study as previously reported by UNICEF [22]. The protective effects of education and wealth status could be because women who have attained high levels of education are more exposed and have access to knowledge and information for making demands for social change to improve their standard of living and support better management of interpersonal relationships [23]. Additionally, educated women are more capable of rebelling against oppressive customs and IPV, hence, the risk of such women becoming victims of IPV is low. Similar findings showing that women from households in the higher wealth spectrum (richer or richest) were less likely to experience IPV and to be underweight is in line with findings from a study conducted in Bangladesh [24]. Also, the findings of the present study indicating that women who live in urban areas, those who have attained secondary education, those from middle wealth quintile households, working women and those who read newspaper and watch television, are less likely to simultaneously suffer from IPV and underweight are consistent with similar findings in Uganda and elsewhere [25–29]. These findings emphasize the role higher education, high

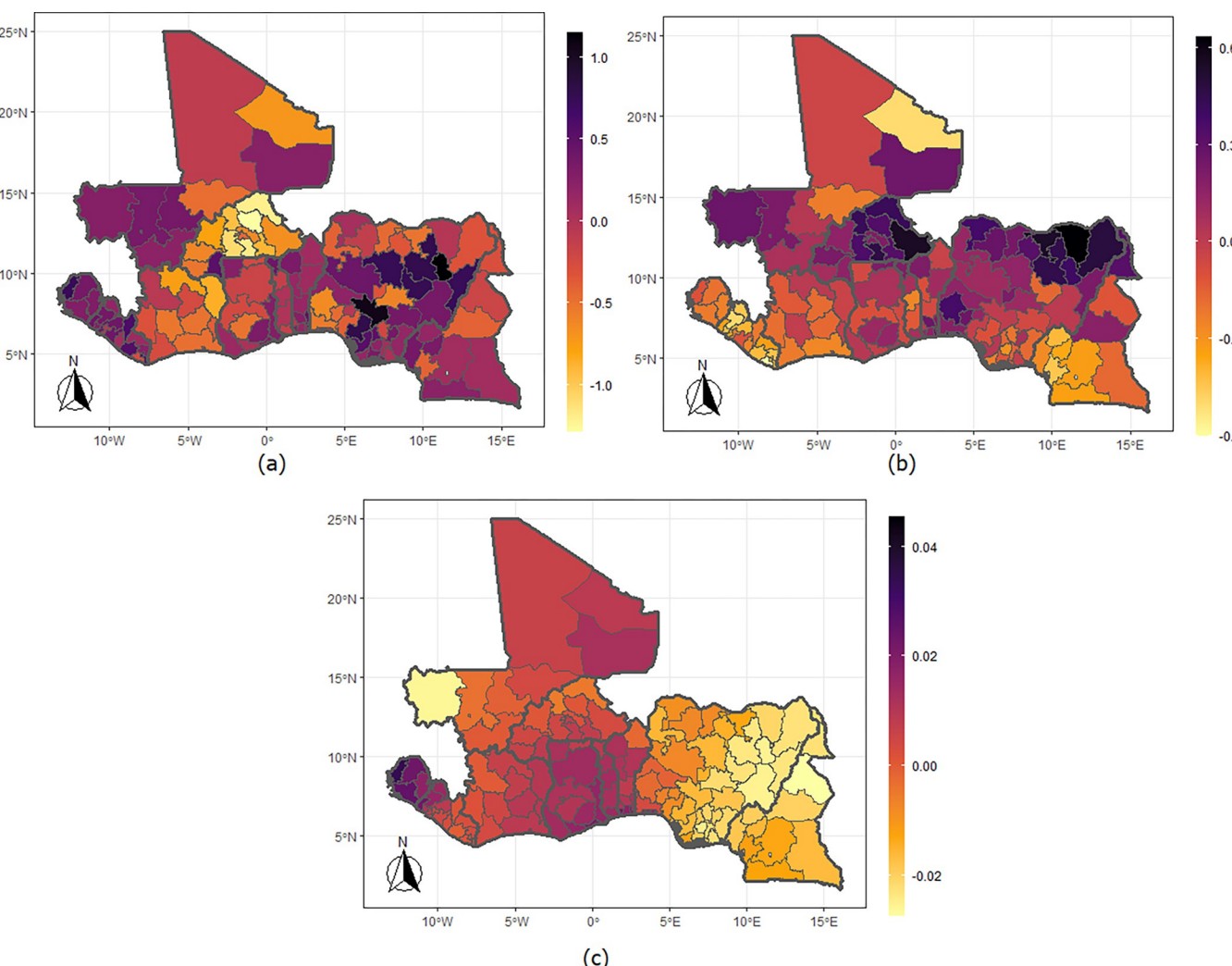

**Fig 4.** Maps of the West African countries showing the mean levels for (a) emotional violence, (b) underweight, and (c) correlation between emotional violence and underweight. Shapefiles used in plotting the maps were sourced from https://spatialdata.dhsprogram.com/home/.

levels of socio-economic status and access to mass media play in empowering women and protecting them against human rights violations such as IPV. Being empowered also means that such categories of women can make their own decisions regarding their health including the choice of food to eat and they can also afford to purchase safe, quality, and nutritious foods for themselves and their children, thus reducing the chances of undernutrition.

The findings of the present study also revealed a negative linear relationship between the experience of IPV and underweight among women who live in urban areas, those who have attained secondary or tertiary education, those from middle wealth status households, those are working, women who read newspaper and those who watch television. The findings, however, indicate that these categories of women, are less likely to be underweight even if they are more likely to experience IPV. The lower likelihood of the simultaneous occurrence of IPV and underweight among these women could be due to the complexity of the interplay of different factors acting together to either protect or expose women to IPV and undernutrition. For women living in urban areas, greater autonomy [30] and the lesser influence of cultural norms and expectations around gender roles in the urban environment provides an escape from vices such as IPV. This notwithstanding,

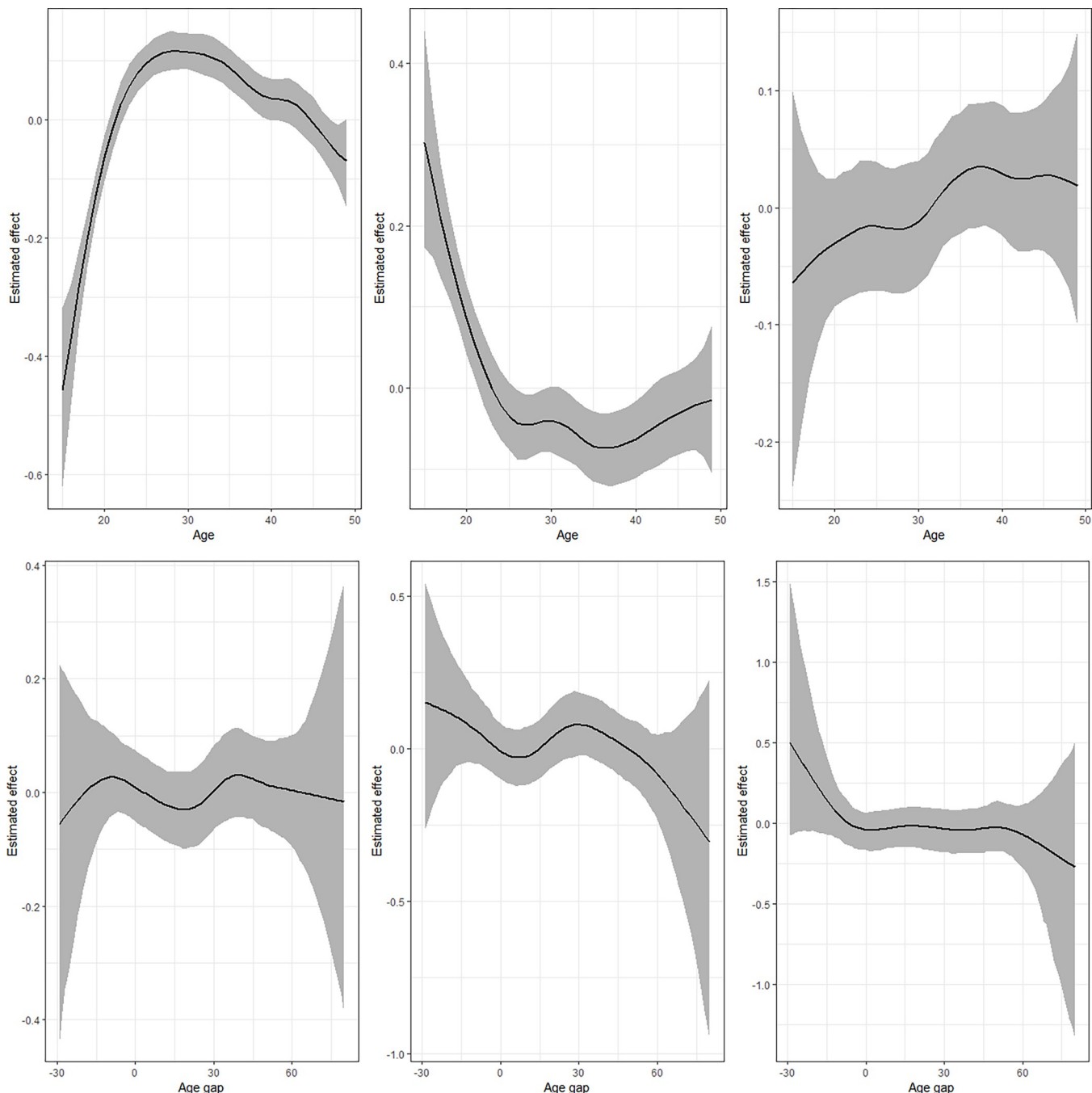

**Fig 5.** Nonlinear effects of age (upper panel) and age gap (lower panel) for the model combining IPV and underweight. Left panel presents estimates for IPV ($\mu_1$), middle is for underweight ($\mu_2$) while left panel is for the correlation parameter ($\rho$).

women in some settings in the urban environment, particularly informal settlements are at higher risk of experiencing IPV [31, 32] compared to those in the affluent urban setting or those in rural areas. Women who live in informal or slum settlements are also often less educated and of lower socio-economic status, thus being less educated and or of poorer social economic status erodes any urban advantage, making them even more vulnerable to IPV. The socio-economic hardships of such environments also put women at risk of undernutrition.

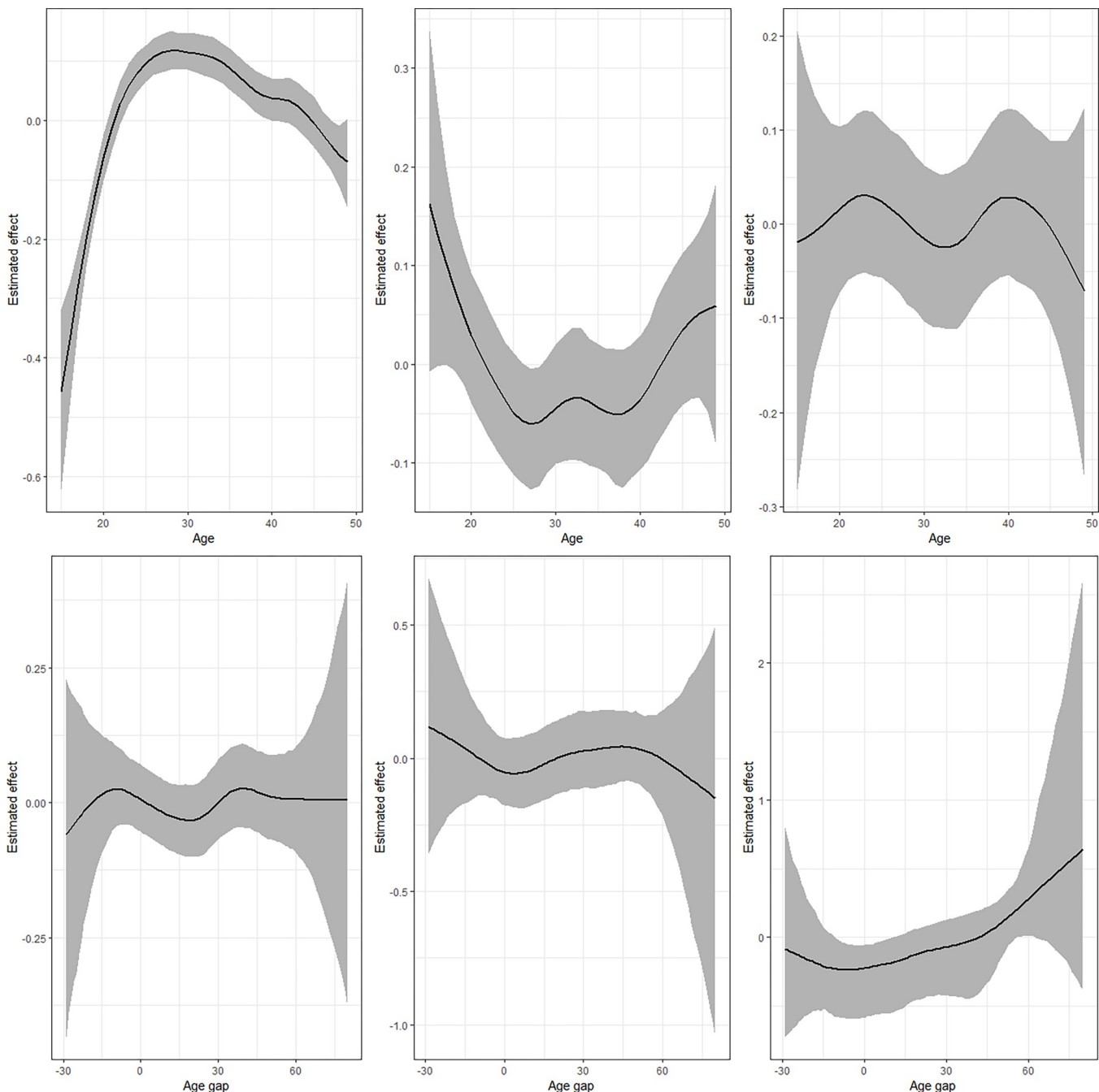

**Fig 6.** Nonlinear effects of age (upper panel) and age gap (lower panel) for the model combining IPV and thinness. Left panel presents estimates for IPV ($\mu_1$), middle is for underweight ($\mu_2$) while left panel is for the correlation parameter ($\rho$).

The higher likelihood of IPV and thinness found among women who read newspaper and those from poorer households and lesser chances for those who attained secondary education, those from the middle households and the working women are supported by findings from other studies [24, 33, 34]. Wealth status has been found to moderate the prevalence of chronic malnutrition [24], hence women from middle wealth status households and those who are working may be more financially stable and may not necessarily become thin due to IPV. This

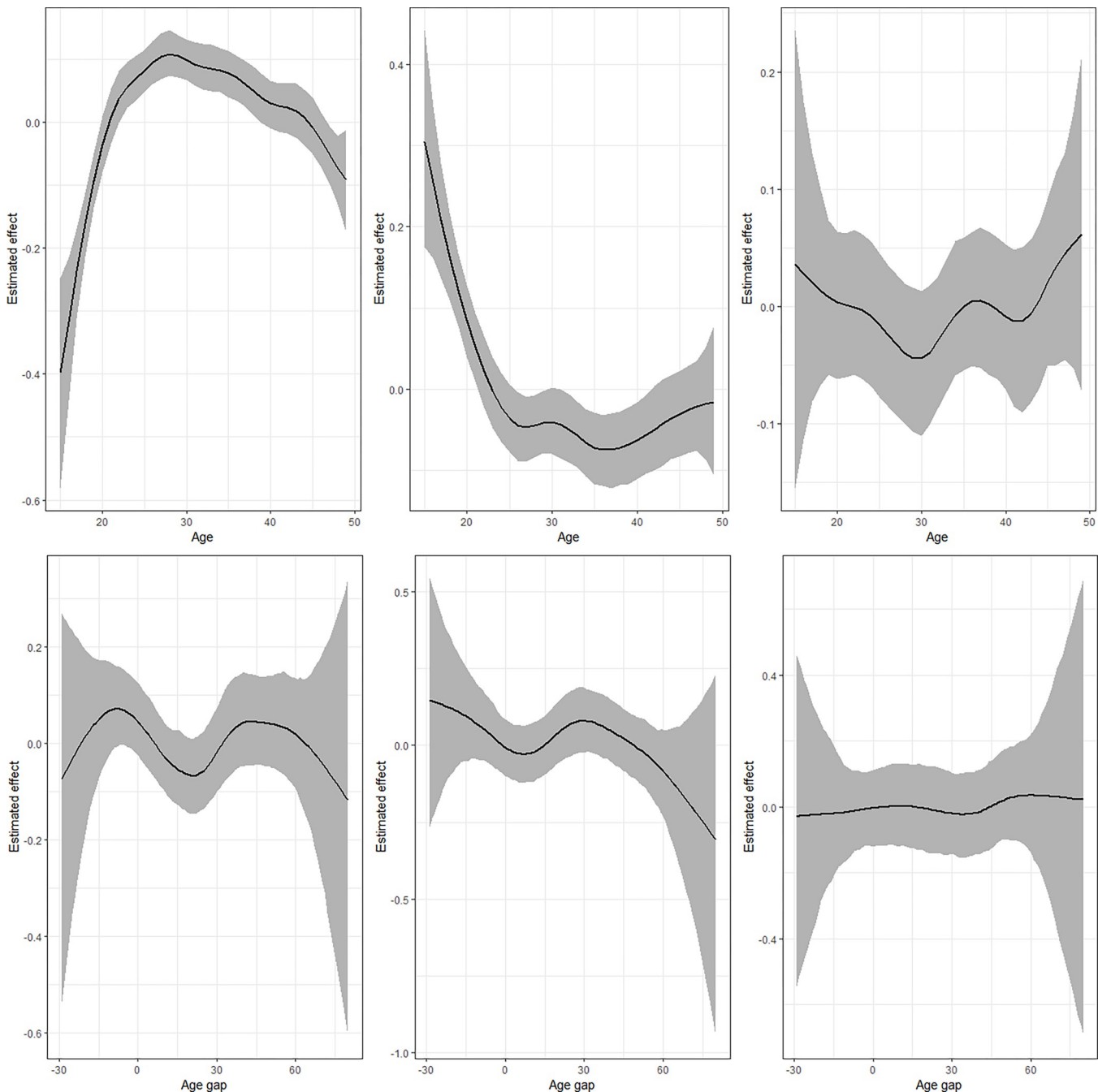

**Fig 7.** Nonlinear effects of age (upper panel) and age gap (lower panel) for the model combining physical violence and underweight. Left panel presents estimates for IPV ($\mu_1$), middle is for underweight ($\mu_2$) while left panel is for the correlation parameter ($\rho$).

is because they have the resources, and in most cases have access to food and can decide what to eat, thereby reducing the risk of being underweight.

The results of the spatial models show strong spatial clustering in IPV and undernutrition within and across the West African region. Some studies have shown spatial clustering in IPV in sub-Saharan Africa [35, 36] Nigeria [37–41], Ghana [42], Ethiopia [30, 43] and South Africa [44] but very few studies have explored the simultaneous occurrence of IPV and

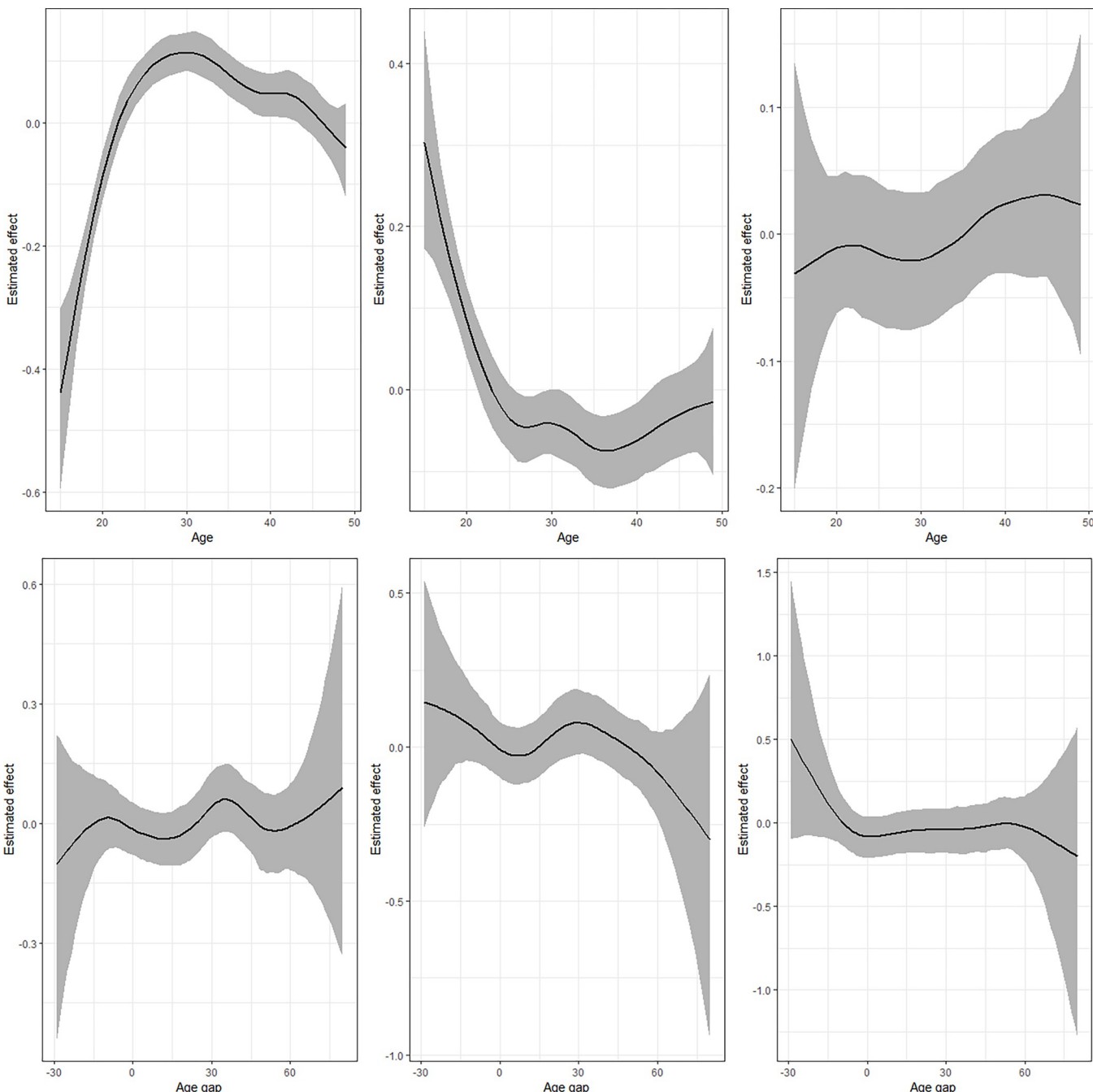

**Fig 8.** Nonlinear effects of age (upper panel) and age gap (lower panel) for the model combining emotional violence and underweight. Left panel presents estimates for IPV ($\mu_1$), middle is for underweight ($\mu_2$) while left panel is for the correlation parameter ($\rho$).

undernutrition in similar geographic spaces. The positive linear relationship between IPV and underweight among women in some regions of Mali; all parts of Sierra Leone and Liberia and neighboring Cote d'Ivoire, Ghana, and Togo as was found in the present study could be because of common contextual/place-based socio-cultural beliefs and practices that perpetuate male dominance and limits women autonomy thus exposing women in these regions to the risk of IPV and undernutrition. For example, justification of IPV has been found to be highest

in Mali [36] and other risk factors such as polygyny and early marriage are also prevalent in Mali [45]. The combination of high justification of IPV, widespread polygyny and early marriage maybe driving the relationship between IPV and undernutrition in Mali as has been found in the current study. In addition, this finding may be influenced by extraneous factors like poverty and geographic location. Mali is one of the poorest countries in the world and geographically located in the Sahel, which consequently may exacerbate the problem of food insecurity and malnutrition. Additionally, in countries such as Nigeria, patriarchal social system that favor male superiority and female subjugation may drive the high levels of IPV [46]. Besides, religious practices may be a probable explanation for the high occurrence of IPV in Nigeria and neighboring country of Benin. For instance, the Islamic practices of early marriage and seclusion of women (Pudah) which are common in Northern Nigeria and all parts of Benin may encourage women to stay with violent partners, as divorce is frowned upon and only considered when all options have been exhausted. Findings from other studies conducted in Ghana also suggest that, the spatial distribution of IPV is not random [42]. This may be attributed to the culture of male superiority and entitlement to sex, expressed through 'control of women and sexual prowess, and payment of bride price which is still practiced across ethnic groups in Ghana [23].

The final set of results, showing the non-linear effects of age of women and age gap and IPV, reflect the expected relationship among these variables. Young women are more likely to be both underweight, and thin and experience IPV compared to older women. These differences are evident in the results for age gap. This result is in line with studies by Haque, Choudhury [47] and Kishor and Johnson (2005) [48]. The likely explanation could be that **as** women age, their socioeconomic standing increases in terms of educational attainment, decision-making role, and the household's wealth and as such, their likelihood of experiencing IPV, being underweight or thin declines. Moreso, this could be indicative of women's changing position in the family as mothers which accords them a more significant position and status than wives in the household and community at large.

## Strengths and limitations of the study

This study uses data from nationally representative survey, and results can be generalized, and the findings are comparable across countries. There are, however, some limitations that are also worth noting. First, the study used cross-sectional data, and therefore cannot establish cause and effects based on associations between malnutrition and women's IPV experienced in the past 12 months. Second, IPV experience may have been underreported in the study countries due to social desirability bias, fear of further abuse by partners, stigmatization or the culture of silence [23].

## Conclusion

The aforementioned limitations notwithstanding, this study has established the prevalence of IPV among women in the West African region and its association with underweight and thinness. Clustering of IPV was found across the various study countries with the highest clustering of IPV and underweight occurring in the northern and western districts of Mali, northern states of Nigeria, southern part of Cameroon, Western and Eastern regions of Ghana most part of Benin, and anywhere in Sierra Leone and Liberia. Key factors identified to be associated with the risk of experiencing IPV include high educational attainment, employment status, household wealth quintile and age. The findings of the study underscore the importance of empowering women holistically, in the domains of education, socio-economic and socio-cultural empowerment, as the combined effect of these factors reduces vulnerability to IPV and

undernourishment as was found in the current study. Against the foregoing, strategies aimed at reducing IPV and malnutrition (underweight, thinness), need to consider empowering women educationally, socially and financially by providing more job opportunities to enable them improve their nutritional status. Furthermore, IPV prevention programmes will need to address gender inequality and cultural factors such as male dominance that may heighten women's IPV risk in general and particularly employed women.

## Supporting information

**S1 Text. Questions used by Demographic and Health Survey to elicit information on each type of intimate partner violence.**
(DOCX)

## Author Contributions

**Conceptualization:** Ezra Gayawan, Dorothy N. Ononokpono.

**Data curation:** Ezra Gayawan, Olabimpe B. Aladeniyi.

**Formal analysis:** Ezra Gayawan.

**Investigation:** Endurance Uzobo, Fidelia A. A. Dake.

**Methodology:** Ezra Gayawan.

**Project administration:** Dorothy N. Ononokpono.

**Supervision:** Ezra Gayawan.

**Writing – original draft:** Ezra Gayawan, Endurance Uzobo, Dorothy N. Ononokpono, Olabimpe B. Aladeniyi, Fidelia A. A. Dake.

**Writing – review & editing:** Ezra Gayawan, Endurance Uzobo, Dorothy N. Ononokpono, Fidelia A. A. Dake.

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
