## [Decision Letter · Decision Letter 0]

9 Jul 2023

PGPH-D-23-00645

Intimate partner violence and malnutrition among women of reproductive age in Western Africa: A geostatistical analysis

Dear Dr. Ezra Gayawan,

Thank you for submitting your manuscript to PLOS Global Public Health. After careful consideration, we feel that it has merit but does not fully meet PLOS Global Public Health’s publication criteria as it currently stands. Therefore, we invite you to submit a revised version of the manuscript that addresses the points raised during the review process.

We look forward to receiving your revised manuscript.

Kind regards,

Jayanta Kumar Bora,PhD

Academic Editor

Journal Requirements:

1. Please note that your Data Availability Statement is currently missing a direct link to access each database. If your manuscript is accepted for publication, you will be asked to provide these details on a very short timeline. We therefore suggest that you provide this information now, though we will not hold up the peer review process if you are unable.

2. Please send a completed 'Competing Interests' statement, including any COIs declared by your co-authors. If you have no competing interests to declare, please state "The authors have declared that no competing interests exist". 

3. Please provide separate figure files in .tif or .eps format and remove the embedded figures from the manuscript file.

4.  Some material included in your submission may be copyrighted. According to PLOS’s copyright policy, authors who use figures or other material (e.g., graphics, clipart, maps) from another author or copyright holder must demonstrate or obtain permission to publish this material under the Creative Commons Attribution 4.0 International (CC BY 4.0) License used by PLOS journals. Please closely review the details of PLOS’s copyright requirements here: PLOS Licenses and Copyright. If you need to request permissions from a copyright holder, you may use PLOS's Copyright Content Permission form.

Potential Copyright Issues:

Figures 1 to 4: please (a) provide a direct link to the base layer of the map (i.e., the country or region border shape) and ensure this is also included in the figure legend; and (b) provide a link to the terms of use / license information for the base layer image or shapefile. We cannot publish proprietary or copyrighted maps (e.g. Google Maps, Mapquest) and the terms of use for your map base layer must be compatible with our CC-BY 4.0 license. 

Additional Editor Comments (if provided):

Reviewers' comments:

Reviewer's Responses to Questions

**Comments to the Author**

1. Does this manuscript meet PLOS Global Public Health’s publication criteria? Is the manuscript technically sound, and do the data support the conclusions? The manuscript must describe methodologically and ethically rigorous research with conclusions that are appropriately drawn based on the data presented.

Reviewer #1: Partly

Reviewer #2: Partly

2. Has the statistical analysis been performed appropriately and rigorously?

Reviewer #1: Yes

Reviewer #2: Yes

3. Have the authors made all data underlying the findings in their manuscript fully available (please refer to the Data Availability Statement at the start of the manuscript PDF file)?

Reviewer #1: Yes

Reviewer #2: Yes

4. Is the manuscript presented in an intelligible fashion and written in standard English?

Reviewer #1: No

Reviewer #2: Yes

5. Review Comments to the Author

Reviewer #1: 1. Thank you for inviting to review this article but my concern is it is not following scientific background to write an article which will be published in reputable journal like PLOS Global Public Health. Abstract of a manuscript which includes at least, Background, objective, method, result, and conclusion. But this is not shows what it follows, and the abstract of this manuscript is not explanatory and gives sound for any one to grasp the overall the manuscript in its abstract.

2. Online “59” Introduction part next to reference 7, there is Harvard style reference. Why it is why not be consistent and be Vancouver style like others?

3. it is better to put statistical measures result which have an association on abstract result and discussion, because it is not possible to know which variable affects positively and negatively the association of intimate partner violence and malnutrition

Reviewer #2: Reviewer’s comments

Abstract section

1. The abstract is too short and does not contain all relevant information that it should contain as per the standard of PLOS ONE. Therefore, this issue must be addressed.

Background section

Well done but the following comment should be considered

1. The introduction is too long because it contains many ideas that are not directly related to the research topic. So, it should focus on the study topic.

2. The research objective and the research contribution to the world are not explained.

Method section

Well done but the following comment should be considered

1. In this section the software used for data analysis is not mentioned.

2. The statistical description method is not indicated

3. The data source section is unnecessarily too long as it contains many irrelevant information such as how DHIS Data were collected and analyzed. So, shorten it.

4. Justification should be given why the authors prefer to use bi-probit and Bayesian approach.

Result section

1. In this section the magnitude or coefficients of the variable s that were compared along with their respective confidence or correlation interval are not shown.

2. The section is congested by presenting all results in one paragraph that makes the result section boring. Thus, to make it attractive the result section should be revisited and each table and figure should be presented after narration.

3. Furthermore, this section like the introduction section is too long. So the section should focus on

Discussion section

This section should be revisited by looking at the following comments

1. It is too long so it should be shortened by focusing on the important points as I commented on result section.

2. The interpretations of results are not clear for reader so that the authors should have to interpret results and compare with the relevant literatures in a way it is understandable.

3. The interpretation should lead to the conclusion and recommendations. Also, it is a must to make sure result answer the research questions and achieve the research objectives.

Conclusion Section

The conclusion section should be revisited again as per the comments given above and in the research word document.

6. PLOS authors have the option to publish the peer review history of their article (what does this mean?). If published, this will include your full peer review and any attached files.

**Do you want your identity to be public for this peer review?** For information about this choice, including consent withdrawal, please see our Privacy Policy.

Reviewer #1: No

Reviewer #2: No

---

## [Decision Letter · Decision Letter 1]

25 Sep 2023

PGPH-D-23-00645R1

Intimate partner violence and malnutrition among women of reproductive age in Western Africa: A geostatistical analysis

Dear Dr. Ezra Gayawan,

Thank you for submitting your manuscript to PLOS Global Public Health. After careful consideration, we feel that it has merit but does not fully meet PLOS Global Public Health’s publication criteria as it currently stands. Therefore, we invite you to submit a revised version of the manuscript that addresses the points raised during the review process.

We look forward to receiving your revised manuscript.

Kind regards,

Jayanta Kumar Bora,PhD

Academic Editor

Journal Requirements:

2. Please provide separate figure files in .tif or .eps format only and remove any figures embedded in your manuscript file. Please also ensure all files are under our size limit of 10MB. For more information about figure files please see our guidelines: https://journals.plos.org/globalpublichealth/s/figures https://journals.plos.org/globalpublichealth/s/figures#loc-file-requirement 3. We have noticed that you have uploaded Supporting Information files, but you have not included a list of legends. Please add 
a full list of legends for your Supporting Information files after the references list. 

Additional Editor Comments (if provided):

Reviewers' comments:

Reviewer's Responses to Questions

**Comments to the Author**

1. If the authors have adequately addressed your comments raised in a previous round of review and you feel that this manuscript is now acceptable for publication, you may indicate that here to bypass the “Comments to the Author” section, enter your conflict of interest statement in the “Confidential to Editor” section, and submit your "Accept" recommendation.

Reviewer #1: All comments have been addressed

Reviewer #2: All comments have been addressed

2. Does this manuscript meet PLOS Global Public Health’s publication criteria? Is the manuscript technically sound, and do the data support the conclusions? The manuscript must describe methodologically and ethically rigorous research with conclusions that are appropriately drawn based on the data presented.

Reviewer #1: Yes

Reviewer #2: Yes

3. Has the statistical analysis been performed appropriately and rigorously?

Reviewer #1: Yes

Reviewer #2: Yes

4. Have the authors made all data underlying the findings in their manuscript fully available (please refer to the Data Availability Statement at the start of the manuscript PDF file)?

Reviewer #1: Yes

Reviewer #2: Yes

5. Is the manuscript presented in an intelligible fashion and written in standard English?

Reviewer #1: Yes

Reviewer #2: Yes

6. Review Comments to the Author

Reviewer #1: Thank you for incorporating the comments and suggestion i put on this manuscript . it was interesting and valuable for scientific community especially the relation on intimeaye partiner violence and malnutrition

withregards!

Reviewer #2: I kindly suggest that the author should address the minor comments that I indicated in the attached comment paper.

7. PLOS authors have the option to publish the peer review history of their article (what does this mean?). If published, this will include your full peer review and any attached files.

**Do you want your identity to be public for this peer review?** For information about this choice, including consent withdrawal, please see our Privacy Policy.

Reviewer #1: No

Reviewer #2: **Yes: **Tamiru Demeke

---

## [Editor Report · Decision Letter 2]

17 Oct 2023

Intimate partner violence and malnutrition among women of reproductive age in Western Africa: A geostatistical analysis

PGPH-D-23-00645R2

Dear Dr. Ezra Gayawan,

We are pleased to inform you that your manuscript 'Intimate partner violence and malnutrition among women of reproductive age in Western Africa: A geostatistical analysis' has been provisionally accepted for publication in PLOS Global Public Health.

Best regards,

Jayanta Kumar Bora,PhD

Academic Editor